# Uridine Prevents Negative Effects of OXPHOS Xenobiotics on Dopaminergic Neuronal Differentiation

**DOI:** 10.3390/cells8111407

**Published:** 2019-11-08

**Authors:** Eldris Iglesias, M. Pilar Bayona-Bafaluy, Alba Pesini, Nuria Garrido-Pérez, Patricia Meade, Paula Gaudó, Irene Jiménez-Salvador, Julio Montoya, Eduardo Ruiz-Pesini

**Affiliations:** 1Departamento de Bioquímica, Biología Molecular y Celular, Universidad de Zaragoza, C/Miguel Servet 177, 50013 Zaragoza, Spain; eiglesia@unizar.es (E.I.); pbayona@unizar.es (M.P.B.-B.); ngarrido@unizar.es (N.G.-P.); pmeade@unizar.es (P.M.); pgaudo@unizar.es (P.G.); jimsal@unizar.es (I.J.-S.); jmontoya@unizar.es (J.M.); 2Instituto de Investigación Sanitaria (IIS) de Aragón, Av. San Juan Bosco 13, 50009 Zaragoza, Spain; 3Centro de Investigaciones Biomédicas en Red de Enfermedades Raras (CIBERER), Av. Monforte de Lemos 3-5, 28029 Madrid, Spain; 4Fundación ARAID. Av. de Ranillas 1-D, 50018 Zaragoza, Spain

**Keywords:** oxidative phosphorylation, xenobiotics, linezolid, uridine, neuronal differentiation, Parkinson’s disease

## Abstract

Neuronal differentiation appears to be dependent on oxidative phosphorylation capacity. Several drugs inhibit oxidative phosphorylation and might be detrimental for neuronal differentiation. Some pregnant women take these medications during their first weeks of gestation when fetal nervous system is being developed. These treatments might have later negative consequences on the offspring’s health. To analyze a potential negative effect of three widely used medications, we studied in vitro dopaminergic neuronal differentiation of cells exposed to pharmacologic concentrations of azidothymidine for acquired immune deficiency syndrome; linezolid for multidrug-resistant tuberculosis; and atovaquone for malaria. We also analyzed the dopaminergic neuronal differentiation in brains of fetuses from pregnant mice exposed to linezolid. The drugs reduced the in vitro oxidative phosphorylation capacity and dopaminergic neuronal differentiation. This differentiation process does not appear to be affected in the prenatally exposed fetus brain. Nevertheless, the global DNA methylation in fetal brain was significantly altered, perhaps linking an early exposure to a negative effect in older life. Uridine was able to prevent the negative effects on in vitro dopaminergic neuronal differentiation and on in vivo global DNA methylation. Uridine could be used as a protective agent against oxidative phosphorylation-inhibiting pharmaceuticals provided during pregnancy when dopaminergic neuronal differentiation is taking place.

## 1. Introduction

Late-onset Parkinson’s disease (PD) is a medical condition characterized by non-motor and motor signs. Motor signs are essentially due to the loss of dopaminergic neurons in the substantia nigra. These cells differ from less vulnerable neurons by having a higher basal rate of oxidative phosphorylation (OXPHOS) [1]. OXPHOS is the main energy provider to power neuronal activity [2]. Although PD is currently understood as a multietiological and multifactorial condition [3], numerous observations suggest that an OXPHOS defect may be a pathogenic event in many cases of PD [4]. Thus, activities and levels of OXPHOS complexes were found to decrease in patients’ substantia nigra [4]. Thirteen OXPHOS system polypeptides (7, 1 and 3 from respiratory complexes I-CI, III-CIII and IV-CIV (abbreviations of respiratory complex I, III and IV are CI, CIII and CIV as already indicated), respectively, and 2 from ATP synthase-CV), along with 2 rRNAs and 22 tRNAs required for their expression, are encoded by the mitochondrial DNA (mtDNA). Very interestingly, levels of mtDNA deletions and point mutations were higher, and the mtDNA copy number was lower, in substantia nigra neurons of PD patients [5,6,7].

Many studies indicate that OXPHOS function is important, not only for mature neuronal function, but also for neuronal differentiation [8,9]. In fact, OXPHOS-related mutant genes reduce neuronal differentiation [10]. Some chemicals that are foreign to human beings, i.e., xenobiotics, bind mtDNA-encoded proteins or RNAs and affect OXPHOS function. If OXPHOS function were important for neuronal differentiation, these xenobiotics would negatively affect neurogenesis, as shown for CI, CIII, CIV and CV inhibitors [9,11,12,13]. Several OXPHOS inhibitor pesticides affect dopaminergic neurogenesis and increase the risk of PD [14]. However, it is not easy to know when, how long and what concentrations people have been exposed to these xenobiotics. On the other hand, this information is usually available for other OXPHOS xenobiotics, such as some pharmaceutical drugs, although unconscious exposures are also possible [15]. Some of these compounds can secondarily affect the respiratory complex activities, mitochondrial translation or replication and, finally, cell differentiation. In fact, we have previously found that ribosomal antibiotics and nucleoside reverse transcriptase inhibitors (NRTIs) diminish adipocyte differentiation of adipose tissue-derived stem cells (hASCs) [16,17].

It has been proposed that decreased neurogenesis would be a key player for PD [18]. If altered neurogenesis is related to PD and OXPHOS function is critical for neurogenesis, then the exposure to OXPHOS xenobiotics might be a risk factor for developing PD [4]. Here we analyzed the effect on dopaminergic neuronal differentiation of three widely used drugs with side effects on the OXPHOS function. In the search for strategies to reverse their adverse consequences, we found that uridine, a necessary compound for cell proliferation of OXPHOS-dysfunctional cells [19], prevented these negative effects on dopaminergic neuronal differentiation.

## 2. Materials and Methods

### 2.1. Cells, Mice and Materials

The neuroblastoma SH-SY5Y cell line with mtDNA (rho^+^ cells) was obtained from Sigma (St. Louis, MO, USA) (catalogue number 94030304, lot 13C014, P 17). The neuroblastoma SH-SY5Y rho^0^ cells, without mtDNA, were a gift of Dr. Anne Chomyn from Prof. Attardi’s laboratory at the California Institute of Technology. Another neuroblastoma SH-SY5Y rho^0^ cell line was generated in our laboratory by 3 months exposure to 10 μg/mL ethidium bromide [20]. Human neural stem cells (hNSCs), H9-derived, were from (Gibco^®^, Life Technologies^TM^, Carlsbad, CA, USA).

Seven-week-old C57BL/6J mice were purchased from Charles River Laboratories. Mice were housed in standard polypropylene cages and maintained under controlled room temperature and humidity with 12:12 h light-dark cycle. Food and water were available ad libitum.

Dulbecco’s Modified Eagle’s Medium (DMEM), DMEM/F-12 medium, fetal bovine serum (FBS), StemPro Neural Supplement, recombinant human fibroblast growth factor-basic (FGFb), recombinant human epidermal growth factor (EGF), L-glutamine, CellStart substrate, and phosphate buffered saline (PBS) were from (Gibco^®^, Life Technologies^TM^, Carlsbad, CA, USA).

Retinoic acid (RA), phorbol 12-myristate 13-acetate (TPA), uridine, atovaquone (ATO), azidothymidine (AZT), linezolid (LIN), 4′,6-diamidino-2-phenylindole (DAPI), paraformaldehyde and Triton X-100 were acquired from (Sigma-Aldrich, St. Louis, MO, USA).

Polymerase chain reaction (PCR) primers, TaqMan^®^ gene expression assays, TaqMan^TM^ MicroRNA Reverse Transcription Kit and TaqMan^®^ miRNA assays were from (Thermo Fisher Scientific, Waltham, MA, USA). QuikChange^TM^ Site-Directed Mutagenesis Kit was obtained from (Stratagene, CA, USA). Direct-zol^TM^ RNA MiniPrep was from (Zymo Research, Irvine, CA, USA).

Rabbit anti-POLG (mitochondrial DNA polymerase gamma), rabbit anti-UQCRFS1 (ubiquinol-cytochrome c reductase, Rieske iron-sulfur polypeptide 1), mouse anti-p.MT-CO1 (OxPhos CIV subunit I), rabbit anti-SDHA (succinate dehydrogenase A) and secondary antibodies DyLight^TM^ were from (Thermo Fisher Scientific, Waltham, MA, USA). Rabbit anti-MRPS12 (mitochondrial ribosomal protein S12), rabbit anti-TH (tyrosine hydroxylase), mouse anti-β-ACTIN, rabbit anti-β-ACTIN, rabbit anti-DAT (dopamine transporter), mouse anti-MAP2 (microtubule-associated protein 2) were from (Sigma-Aldrich, St. Louis, MO, USA). Rabbit anti-TUBB3 (βIII-tubulin), rabbit anti-NSE (neuron-specific enolase), rabbit anti-DCX (doublecortin), mouse anti-NES (nestin) and mouse anti-total OXPHOS human WB antibody cocktail were purchased from (Abcam, Cambridge, UK). Super Signal West Pico Chemiluminescence Substrate was from (Thermo Fisher Scientific, Waltham, MA, USA). Alexa Fluor^®^ 488 and 594 were from (Molecular Probes, Eugene, OR, USA).

MethylFlash^TM^ Methylated DNA Quantification Kit (Epigentek, Farmingdale, NY, USA). Dopamine/noradrenaline ELISA kit (Rocky Mountain Diagnostics, Colorado Springs, CO, USA). Amplex Red Acetylcholine/Acetylcholinesterase Assay Kit (Gibco^®^, Life Technologies^TM^, Carlsbad, CA, USA).

### 2.2. Cell Culture and Differentiation

SH-SY5Y cells were cultured in DMEM high glucose (25 mM), 4 mM L-glutamine, and 1 mM sodium pyruvate and supplemented with 10% FBS. SH-SY5Y rho^0^ cells were supplemented with 200 μM uridine. hNSCs (<10 passages) were grown in KnockOut^TM^ DMEM/F-12 medium containing StemPro Neural Supplement (2%), FGFb (20 ng/mL), EGF (20 ng/mL) and L-glutamine (2 mM), prior to attachment on cell culture plates coated with CellStart.

Dopaminergic differentiation of SH-SY5Y cells was induced following optimized protocols [21]. To make cells more dependent on OXPHOS function, SH-SY5Y rho^+^ cells were differentiated in 5 mM galactose media instead of 25 mM glucose. SH-SY5Y rho^0^ cells were differentiated in 25 mM glucose and supplemented with 200 μM uridine. Dopaminergic differentiation of hNSCs was performed as previously described [22].

Cells were maintained at 37 °C in a humidified atmosphere of 5% CO_2_. Cell number was determined with trypan blue stain using a Countess Automated Cell Counter (Invitrogen, Thermo Fisher Scientific, Waltham, MA, USA).

### 2.3. Pharmacological Treatments

For neuronal differentiation in the presence of xenobiotics, SH-SY5Y cells were exposed to AZT (5 μM), LIN (40 μM) or ATO (0.5 μM). These drugs were dissolved in ethanol 100%. Final ethanol concentration in the media did not exceed 0.025%.

AZT is used to treat the acquired immune deficiency syndrome (AIDS) because it inhibits the human immunodeficiency virus (HIV) reverse transcriptase. However, it also affects human POLG, and mtDNA replication. Thus, OXPHOS function is a common off-target of NRTIs toxicity [23]. For AZT, the current recommendations for adult patients call for 300 mg every 12 h. This dose is approximately equivalent to 10 mg/kg/day, and results in a steady-state serum AZT concentration of 0.8 μM and peak concentrations (Cmax) of around 5 μM AZT [24].

LIN is a ribosomal antibiotic used in the treatment of multidrug-resistant tuberculosis (MDR-TB) because inhibits the bacterial ribosome. However, it also inhibits mitochondrial protein translation. The protein synthesis apparatus of mitochondria is similar to that of bacteria because of its endosymbiotic origin and, therefore, mitochondrial ribosomes are frequently unintended off-targets of these ribosomal antibiotics used in clinical practice to fight pathogenic bacteria. These antibiotics can provoke serious adverse drug reactions in patients [25]. The LIN dose used in clinical practice is 600 mg every 12 h. This dose results in steady-state peak serum concentrations of 44.5–80 μM [26].

ATO is used against *Plasmodium falciparum*, the parasite that causes malaria, and other parasites causing opportunistic AIDS infections. Interspecies differences in the drug target, the coenzyme Q outer pocket (Qo site), have allowed its use against the parasites with few adverse effects on humans. We had previously proposed that genetic variation in the human p.MT-CYB subunit from CIII would generate structurally different Qo sites that also affect xenobiotics binding [27]. In fact, it has been recently reported that a yeast model harboring a human haplogroup-defining Qo site variant increased ATO sensitivity [28]. Moreover, this drug also inhibited CIII activity and oxygen consumption in human cell lines [29]. The ATO dose used in clinical practice is 750 mg every 12 h. This dose generates average blood concentrations around 40 μM [30]. We chose a low concentration because we noted that concentrations higher than 1 μM had a very important effect on oxygen consumption of SH-SY5Y cells.

### 2.4. Cell Proliferation

SH-SY5Y cells were seeded on 100 mm plates (1 × 10^5^ cells) and treated with AZT, LIN or ATO at concentrations of 5, 10 and 0.5 µM, respectively. Cells were exposed to drugs for 7 days. Then, whole cell lysates were obtained using radioimmunoprecipitation assay (RIPA) buffer. Protein concentration was determined by the Bradford protocol (Bio-Rad, Hercules, CA, USA).

### 2.5. Overexpressing OXPHOS-Related Mutant Proteins

*POLG*, *MRPS12* and *UQCRFS1* constructs were obtained and introduced in the SH-SY5Y cells using a lentiviral system [31]. We chose these proteins because they participate in the same mitochondrial processes than the previously cited OXPHOS xenobiotics (replication—POLG and AZT, translation—MRPS12 and LIN, and respiratory chain function—UQCRSF1 and ATO). *POLG* (RefSeq NM_002693; NP_002684), *MRPS12* (RefSeq Variant 1 NM_021107.1; NP_066930.1) and *UQCRFS1* (RefSeq NM_006003; NP_005994) were PCR amplified with following primers:

*POLG* Fw: GTTTAAACGCCACCATGAGCCGCCTGCTCT and Rv: GGATCCCTATGGTCCA GGCTGG;

*MRPS12* Fw: GTTTAAACGCCACCATGTCCTGGTCTGGCC and Rv: GTTTAAACTGTTTA TTAAAACCCC;

*UQCRFS1* Fw: GTTTAAACGCCACCATGTTGTCGGTAGCATCCCG and Rv: GGATCCTT AACCAACAATCACCATATCGTCACTGG.

A sequence checked clone was used as template for site directed mutagenesis by using QuikChange^®^ Site-Directed Mutagenesis Kit and the mutagenic primers following:

*POLG* Fw: CTACGGCCGCATCTGTGGTGCTGGGCAGC and Rv: GCTGCCCAGCACCACAGATGCGGCCGTAG; *MRPS12* Fw: CTGTGCACGTTTACCCTCAAGCCGAAGAAGCC and Rv: GGCTTC TTCGGCTTGAGGGTAAACGTGCACAG and *UQCRFS1* Fw: GCACTCATCTTGGCTCTG TACCCATTGCAAATGC and Rv: CGTGAGTAGAACCGAGACATGGGTAACGTTTACG.

Overexpressed variants of these genes were sequenced from retro-transcribed cellular RNA with the same primers used for cloning.

### 2.6. Chromosomes and Mitochondrial DNA Analysis

Nuclear genetic fingerprint, karyotyping, mtDNA sequencing and mtDNA levels were determined according to protocols previously reported [16,32]. For mtDNA sequencing, long-PCR reactions were carried out in 50 µL reaction mixture containing 25 µL of 2X Phusion Master Mix with GC Buffer (Thermo Fisher Scientific), 1 µL (0.5 µM) of each primer (*MT-ND2-L*: 5′-TTAATCCCCTGGCCCAACCCGTCATCTACTC-3′ and *MT-ND2-H*: 5′-CGGATACAGTTCAC TTTAGCTACCCCCAAGTG-3′), 22 µL of sterile distilled H_2_O and DNA template (100 ng). The reaction consisted of an initial denaturing step at 98 °C for 30 s, followed by 30 cycles of denaturing at 98 °C for 10 s, extension at 72 °C for 8 min 15 s, and a final extension at 72 °C for 10 min.

### 2.7. DNA Methylation Analysis

For quantitative determination of 5-methylcytosine (5-mC) percentage in genome of mouse brain samples, the MethylFlash^TM^ Methylated DNA Quantification Kit was used following the manufacturer’s instructions.

### 2.8. RT-qPCR Analysis

*POLG*, *MRPS12* and *UQCRFS1* mRNA levels were determined, in SH-SY5Y cells, by quantitative PCR assays that were carried out in a LightCycler 2.0 system (Roche), using FastStart DNA MasterPLUS SYBR Green I (Roche) and primers qMRPS12-36 Fw: AGGCAGCCACTCATGGATT, qMRPS12-36 Rv: GGCTTAATAGTGGTCCTGATGG, qPOLG#5 Fw: ACGCCCATAAACGTATCAGC, qPOLG#5 Rv: CATAGTCGGGGTGCCTGA, qUQCRFS1#30 Fw: CCTGTGTTGGACCTGAAGC and qUQCRFS1#30 Rv: ATAACAAACAGAAGCAGGGACAT, respectively. The mRNA levels of subunits 2 and 6 (*MT-ND2*, *MT-ND6*) from CI, of peroxisome proliferator-activated receptor gamma (*PPARG*), and *12S* rRNA amount were determined and normalized using the *18S* rRNA levels [16,32].

Total RNA, including microRNA, was isolated from the whole brain of each embryo using the Direct-zol^TM^ RNA MiniPrep according to the manufacturer’s instructions. Thirty ng of RNAs were used for reverse transcription using the TaqMan^TM^ MicroRNA Reverse Transcription Kit following the manufacturer’s instructions. Relative quantification of mRNA expression was performed by TaqMan real-time PCR using the commercial probes described below, according to the manufacturer’s protocol. Probes were as follows: Engrailed-1, *En 1* (Mm00438709_m1); paired-like homeodomain transcription factor 3, *Pitx3* (Mm01194166_g1); and nuclear receptor-related 1, *NURR1* (or *Nr4a2*) (Mm00443060_m1) and human glyceraldehyde-3-phosphate dehydrogenase (*GAPDH*) (Mm99999915_g1). The samples were normalized with *GAPDH*. Individual TaqMan^®^ miRNA assays were used to quantify the mouse brain samples-expressed miRNAs (Mature miRNA Sequences: *mmu-miR-124a-3p*, UAAGGCACGCGGUGAAUGCC; *mmu-miR-132-3p*, UAACAGUCUACAGCCAUGGUCG; *mmu-miR-133b-3p*, UUUGGUCCCCUUCAACCAGCUA). The mouse brain samples were normalized with snoRNA202.

### 2.9. Western Blotting

Western blot (WB) was used to determine protein levels in whole-cell lysates obtained using RIPA buffer. Samples were resolved on sodium dodecyl sulfate-polyacrylamide gel electrophoresis (SDS-PAGE) minigels (Miniprotean, Bio-Rad, Hercules, CA, USA) and were transferred to polyvinylidene difluoride (PVDF) membranes (Trans-Blot^®^ Turbo^TM^ Mini PVDF Transfer Pack, Bio-Rad) using a Trans-Blot^®^ Turbo^TM^ Blotting System (Bio-Rad). Membranes were analyzed by immunoblotting with the following antibodies: rabbit anti-POLG (1:500), rabbit anti-MRPS12 (1:2000), rabbit anti-UQCRFS1 (1:1500) and rabbit anti-β-ACTIN (1:4,000), rabbit anti-TUBB3 (1:1000), rabbit anti-NSE (1:1000), rabbit anti-DCX (1 μg/mL), rabbit anti-TH (1:200), rabbit anti-DAT (1:200), mouse anti-MAP2 (1:500), mouse anti-β-ACTIN (1:1000), mouse anti-p.MT-CO1 (1:1000) and rabbit anti-SDHA (1:1000). After washing, the membrane was incubated with peroxidase-conjugated secondary antibodies (1:5000 or 1:10000) for 1 h at room temperature or with appropriate secondary antibodies DyLight^TM^ (1:15000). Bands were visualized with Super Signal West Pico Chemiluminescence Substrate from PIERCE^®^ or using Odyssey^®^ CLx Imaging System (LI-COR Biosciences, Lincoln, NE, USA).

### 2.10. Flow Cytometry

For flow cytometry (FC) analysis, cell suspensions were fixed with 4% paraformaldehyde for 15 min at 4 °C and permeabilized using a commercial buffer (Thermo Fisher Scientific). Samples were incubated overnight with the appropriate primary antibodies: rabbit anti-TUBB3 (1:1000), mouse anti-NES (1:500), rabbit anti-DCX (1μg/mL), rabbit anti-TH (1:200) and rabbit anti-DAT (1:200), then washed with PBS, and incubated for 30 min with appropriate secondary antibodies Alexa Fluor^®^ 488 (1:1000), washed and, then, analyzed on a BD FACScan System (Becton-Dickinson, San Jose, CA, USA). 10,000 cells were examined. The results were analyzed using Weasel software v2.6.1.

### 2.11. Immunocytochemistry

For immunocytochemistry, cells were fixed with 4% paraformaldehyde for 15 min at room temperature and permeabilized using 0.1% Triton X-100 (Sigma-Aldrich) diluted in PBS for 10 min. To block unspecific epitopes, cells were incubated with 0.1% bovine serum albumin. Primary antibodies (anti-TUBB3, 1:1000 and anti-TH, 1:200) were incubated overnight at 4 °C followed by incubation with appropriate fluorescently labeled secondary antibodies, Alexa Fluor^®^ 488 and 594 (1:1000) for 1 h at room temperature. Finally, cell nuclei were counterstained with 4′,6-diamidino-2-phenylindole (DAPI) (Sigma-Aldrich). Image acquisition was performed using a FLoid^TM^ Cell Imaging Station (Thermo Fisher Scientific).

### 2.12. ELISA

Dopamine/noradrenaline and acetylcholine measurements were performed using commercially available dopamine/noradrenaline and Amplex Red Acetylcholine/Acetylcholinesterase ELISA kits. Samples were collected and measured immediately according to manufacturer’s instruction. For the analysis of dopamine/noradrenaline release, cell membrane depolarization was induced with 100 mM KCl for 10 min. KCl-containing medium was replaced with fresh culture medium and cells were incubated for another 6 h. Then, media and cells were collected and analyzed [33]. Protein levels were analyzed in whole-cell lysates obtained using RIPA buffer.

### 2.13. Analysis of Oxidative Phosphorylation Function

Citrate synthase (CS) specific activity, protein amount, CIV specific activity and quantity, oxygen consumption and mitochondrial protein synthesis were measured according to protocols previously published [34,35].

### 2.14. Mouse Experiments

All animal procedures were carried out under Project License PI03/17, approved by the Ethic Committee for Animal Experiments from the Universidad de Zaragoza. The care and use of animals were performed accordingly with the Spanish Policy for Animal Protection RD53/2013, which meets the European Union Directive 2010/63 on the protection of animals used for experimental and other scientific purposes.

As previously mentioned, humans receive 1,200 mg LIN per day. The mouse equivalent dose, calculated according to the body surface normalization method [36], would be 5 mg LIN per day, well below the limit dose generally accepted in studies of toxicity in mice [37]. Between embryonic days E8 and E15 (8 days), 8-week-old C57BL/6J pregnant mice were daily administered, either with vehicle (methyl cellulose 1%), LIN 5 mg/day, uridine 70 mg/day or LIN plus uridine by oral gavage. On embryonic day E16, pregnant dams were sacrificed by cervical dislocation and fetal brains were obtained for further analysis.

### 2.15. Statistical Analysis

The statistical package StatView 5.0 was used to perform all the statistical analyses. Mann–Whitney and Kruskal–Wallis non-parametric tests were used to compare parameters. To compare more than two groups, post-hoc tests were also performed. All data were expressed as mean ± standard deviation and number of independent experiments (N) and significance levels were set at *p* < 0.05 and the levels indicated by the post-hoc tests.

## 3. Results

### 3.1. OXPHOS Function and Neuronal Differentiation

Firstly, we studied the dopaminergic neuronal differentiation of human neuroblastoma SH-SY5Y cells and compared it with that of hNSCs. Neuronal markers similarly increase with differentiation in both cells, and this differentiation process was very specific for dopaminergic neurons (Appendix A and Appendix A).

Then, we analyzed OXPHOS changes along neuronal differentiation. Despite the fact that the changes in dopaminergic neuronal parameters after differentiation were similar in SH-SY5Y cells and hNSCs, we detected large differences in OXPHOS variables after this process (Appendix A). However, substantial differences in mitochondrial parameters after neuronal differentiation had been already reported [8,38], even within SH-SY5Y cells [39,40,41]. Moreover, several studies reported that cells harboring OXPHOS-related mutant genes were able to normally differentiate into neurons [10,42,43,44,45,46].

All these observations raise doubts about the role of OXPHOS in neural differentiation, although these disparities could also be due to methodological differences [47].

### 3.2. OXPHOS Dysfunction and Dopaminergic Neuronal Differentiation

To corroborate the importance of OXPHOS function on dopaminergic neuronal differentiation, we overexpressed OXPHOS-related mutant proteins in neuroblastoma SH-SY5Y cells (Appendix A). Then, we confirmed their effect on OXPHOS function and analyzed their dopaminergic neuronal differentiation ability in similar conditions.

The levels of mtDNA, OXPHOS subunits and oxygen consumption were lower in SH-SY5Y cells overexpressing the mutant POLG than the wild type (Figure 1A–C). Mutant cells showed reduced TUBB3 and TH amount when compared with wild-type cells (Figure 1D).

The levels of OXPHOS subunits and oxygen consumption were lower in SH-SY5Y cells overexpressing the mutant MRPS12 protein than in cells overexpressing the wild-type version (Figure 1E,F). Cells overexpressing the mutant version showed reduced TUBB3 and TH amount when compared with those overexpressing the wild-type version (Figure 1G).

Cells overexpressing the mutant version of UQCRFS1 did not show differences in OXPHOS subunit levels, and only a mild reduction in oxygen consumption levels was found when compared with those overexpressing the wild-type protein (Figure 1H,I). However, mutant cells showed reduced TUBB3 and TH amount when compared with wild-type cells (Figure 1J).

### 3.3. OXPHOS Xenobiotics and Dopaminergic Neuronal Differentiation

It was reported that one PD-inducing xenobiotic, the CI inhibitor 1-methyl-4-phenylpyridinium ion (MPP^+^), also interfered with mtDNA replication by destabilization of the mtDNA D-loop [48]. The inhibition of mtDNA replication, the translation of mtDNA-encoded mRNAs or respiratory complex activities by chemical compounds (AZT, LIN, ATO) might cause similar effects on dopaminergic neuronal differentiation as overexpressing of OXPHOS-related mutant proteins (POLG, MRPS12, UQCRFS1). At the concentrations and times we used, these OXPHOS xenobiotics did not affect the proliferation of SH-SY5Y cells.

**Azidothymidine.** In SH-SY5Y cells, the presence of 5 μM AZT did not decrease mtDNA levels or the OXPHOS complex subunits, but reduced oxygen consumption (Figure 2A–C). Similarly, it was reported that, at higher AZT concentrations, there was no reduction in mtDNA amount in other cultured cells [49]. Moreover, it was published that, at these high AZT concentrations, mtDNA-encoded subunit levels and most respiratory complex activities were not decreased in human hepatoma HepG2 cells [49]. However, it was informed that, in rat H9c2 cardiomyocytes, oxygen consumption was reduced and, in mouse primary cortical neurons, AZT treatment decreased mitochondrial respiration without any effect on mtDNA levels [50,51]. During our differentiation process, the increase in TUBB3 and TH levels was significantly lower in 5-μM AZT-treated cells (Figure 2D). Similar to this result, it has been previously shown that treatment with 10 μM AZT induced a significant decreases of MAP-2 and synaptophysin in rat cerebrocortical cells after 7 days [52].

**Linezolid.** We analyzed the LIN effect on OXPHOS function of SH-SY5Y cells. Mitochondrial protein synthesis and the amount of the p.MT-CO1 subunit from CIV were moderately reduced by 40 μM LIN (Figure 2E,F). However, oxygen consumption was significantly decreased (Figure 2G). Some of these parameters had been earlier reported decreased in LIN treated human osteosarcoma 143B cybrids, hASCs, and adipocyte-differentiated hASCs [16,17,25]. The increase in TUBB3 and TH levels was significantly lower in 40 μM LIN-treated cells (Figure 2H). It was previously shown that, another mitochondrial protein synthesis inhibitor, chloramphenicol prevented neurite outgrowth and synaptophysin increase in C1300 murine neuroblastoma cells [53], reduced neurite length in rat PC12-ND6 cells and the frequency of stage 3 mouse hippocampal neurons [54], and reduced mouse primary cortical neuron differentiation [55].

**Atovaquone.** We also studied the ATO effect on OXPHOS function of these cells. The p.MT-CO1 subunit levels from CIV were not affected by 0.5 μM ATO, but oxygen consumption was significantly reduced (Figure 2I,J). The increase in TUBB3 and TH levels was significantly lower in 0.5 μM ATO-treated cells (Figure 2K).

### 3.4. High Glucose or Uridine Compensate OXPHOS-Dysfunction Effects on Dopaminergic Neuronal Differentiation

All previous results support an important role for OXPHOS function on dopaminergic neuronal differentiation. In order to better understand the studies reporting normal neuronal differentiation in cells harboring OXPHOS-related mutant genes [10,42,43,44,45,46], we have studied a neuroblastoma SH-SY5Y rho^0^ cell line and, therefore, with no OXPHOS function.

**Neuroblastoma SH-SY5Y rho^0^ cell line.** We confirmed the absence of mtDNA in a neuroblastoma SH-SY5Y rho^0^ cell line (Appendix A). As previously commented, rho^0^ cells require high glucose and uridine to survive. These cells with no mtDNA or OXPHOS function were able to differentiate into dopaminergic neurons, as determined by neurite production and increased TUBB3, NES, MAP2, TH and DAT levels (Figure 3A–C). Another neuroblastoma SH-SY5Y rho^0^ cell line generated in our laboratory was also able to differentiate into dopaminergic neurons (data not shown).

If rho^0^ cells, without OXPHOS function, differentiate into dopaminergic neurons, perhaps OXPHOS xenobiotics, such as AZT, LIN and ATO, affect differentiation to dopaminergic neuron through a mechanism not involving OXPHOS function. To check this possibility, we differentiated SH-SY5Y rho^0^ cells in the presence of 40 μM LIN. There were no significant differences between LIN-untreated and treated differentiated rho^0^ cells. They were able to undergo dopaminergic neuronal differentiation (Figure 3B,C). In fact, it was previously shown that the oxazolidinone eperezolid, similar to LIN, caused no growth inhibition of rho^0^ cells, although it affected parental cells [56]. Besides, it had been also reported that ATO suppressed apoptosis in parental cells but not rho^0^ cells [29]. These results confirm that the effect of these drugs was carried out through an inhibition of the OXPHOS function.

**Rho^0^ medium recovers dopaminergic neuronal differentiation of OXPHOS-dysfunctional cells.** Similar to rho^0^ cells, cells overexpressing a mutant MRPS12 protein, 40 μM LIN-treated cells, or cells overexpressing a mutant MRPS12 protein treated with 40 μM LIN were able to differentiate into dopaminergic neurons when the differentiation is induced in rho^0^ medium containing high glucose (25 mM) and uridine (200 μM) (Figure 3D–F).

**High glucose recovers dopaminergic neuronal differentiation of OXPHOS-dysfunctional cells.** As OXPHOS-dysfunctional SH-SY5Y cells were able to differentiate into dopaminergic neurons in differentiation medium supplemented with 25 mM glucose plus 200 μM uridine, we checked if this glucose concentration, without additional uridine, was enough to rescue the neuronal differentiation of cells overexpressing a mutant MRPS12 protein, 40 μM LIN-treated cells, or cells overexpressing a mutant MRPS12 protein treated with 40 μM LIN. Twenty-five mM glucose was able to recover dopaminergic neuronal differentiation of all these cells (Figure 3G–I). However, 25 mM glucose in medium with dialyzed serum was able to slightly increase TUBB3, but not TH, levels of LIN-treated cells (data not shown).

**High uridine recovers dopaminergic neuronal differentiation of OXPHOS-dysfunctional cells.** Rho^0^ cells also need uridine to survive and proliferate [19], and uridine is lost when serum is dialyzed. Possibly uridine was also required for the dopaminergic neuronal differentiation of SH-SY5Y rho^0^ cells.

Two-hundred μM uridine was able to recover dopaminergic neuronal differentiation of cells overexpressing a mutant MRPS12 protein, 40 μM LIN-treated cells, or cells overexpressing a mutant MRPS12 protein treated with 40 μM LIN (Figure 3J–L). In fact, it has been previously shown that chick embryo cells were resistant to the growth-inhibitory effects of another ribosomal antibiotic, chloramphenicol, when cultured in the presence of uridine [57]. It was also reported that, after 4 days of uridine exposure, PC12 rat pheochromocytoma cells differentiated by nerve growth factor significantly and dose-dependently increased the number of neurites per cell. This rise was accompanied by increases in neurite branching and in levels of the neurite proteins neurofilament M and 70. It has also been found that uridine treatment also increases intracellular levels of cytidine triphosphate, which suggests that uridine may affect neurite outgrowth by enhancing phosphatidylcholine synthesis [58,59].

### 3.5. Effect of Prenatal Exposure to Linezolid on Mouse Brain

All the results shown above indicate that OXPHOS function is relevant for dopaminergic neuronal differentiation. This process is part of the neurogenesis, i.e., the birth of new neurons from stem cells. In adult substantia nigra, neurogenesis remains controversial [60]. This process mainly occurs during development of the nervous system in early life [61]. Thus, in humans, dopaminergic neurons from substantia nigra *pars compacta* are generated from the early fifth (5.3) through the middle part of the seventh (7.0) week after fertilization [62]. The specification and differentiation of midbrain dopaminergic neurons in mice occurs between embryonic days 8–15 [62].

We supplied pregnant mice daily between embryonic days E8 and E15 with vehicle, 5 mg LIN, 70 mg uridine or 5 mg LIN plus 70 mg uridine. The fetus brains were analyzed on embryonic day E16. There were no significant differences in CIV activity or quantity; in dopamine concentration; in TH or DAT amount; in *En1*, *Nr4a2* and *Pitx3* mRNA levels; or in miRNA 124, 132 or 133 levels (Appendix A). However, LIN increased global DNA methylation in fetal brain (Figure 4) and, remarkably, uridine was able to reduce it.

The PD incidence peak occurs between the age of 70 to 79 years. This human age is equivalent to 20 months of age in mice [63]. Hence, older mice should be analyzed to appreciate LIN effects on their brain. However, the early effect in methylation we have observed might suggest that the early exposure to OXPHOS xenobiotics could interfere with developmental programming of nigrostriatal neurons [64]. Thus, the long-lasting modification of global DNA methylation levels might be the connection between these two distant life events: prenatal exposure and old age PD manifestation.

## 4. Discussion

As we have shown, some pharmaceutical drugs inhibiting OXPHOS function reduce dopaminergic neuronal differentiation. The exposure to some of these drugs is not rare. For example, NRTIs are used to prevent mother-to-child HIV transmission. Perinatal HIV transmission is virtually zero in mothers who start NRTIs before conception [65]. In fact, the most recent US guidelines for preconception counseling suggest starting NRTIs when planning pregnancy [66]. Nevertheless, it has been shown that pregnant mice exposed to NRTI in doses that correspond to those used for pregnant HIV-positive women significantly decreased the inducible neurogenesis potential [24]. In utero, AZT exposure perturbs neurogenesis among neural stem/progenitor cells and a short-term AZT regimen in adult mice suppresses subependymal zone neurogenesis [24]. Moreover, some children (0.3%) exposed to these analogues during the perinatal period are at risk of a neurological syndrome, including motor abnormalities, associated with persistent OXPHOS dysfunction during the first 18 months [67].

ATO is an effective drug for malaria prophylaxis and treatment that is not currently recommended for use by pregnant women due to insufficient data on its safety in pregnancy. Besides that, the use of anti-malarial in pregnancy is likely to occur before the woman in question knows of her pregnancy [68], even in the first weeks (3 to 8) after conception [69].

Tuberculosis prevalence in pregnant women is substantial [70], and is associated with poor outcomes [71]. Because of that, ideally all pregnant women should start the treatment as soon as possible. LIN is one of the second-line drugs used for treatment MDR-TB in pregnancy. However, according to the US Food and Drug Administration, LIN is a pregnancy class C agent, meaning that animal studies demonstrate a risk and there are no human studies [71]. Here, we show that LIN affects dopaminergic neuronal differentiation of neuroblastoma SH-SY5Y cells, but does not affect these parameters in mouse fetal brains. However, global DNA methylation in fetal brains was altered.

Uridine might complement the administration of these drugs during the first trimester of pregnancy. Uridine inhibited negative effects of LIN on dopaminergic neuronal differentiation and also reduced global DNA methylation. It was previously shown that, in particular conditions, lack of OXPHOS function is not an obstacle for cell differentiation. Thus, a human rhabdomyosarcoma RD rho^0^ cell line retained the ability to differentiate into myotubes with the expression of muscle specific isoenzymes, such as creatine kinase M (CK-M) [72]. Differentiation along the monocyte/macrophage pathway was very pronounced in the HL60 human promyelocytic leukemia rho^0^ cell line [73]. There is another report showing that neuroblastoma SH-SY5Y rho^0^ cells under neural differentiation protocol expressed long neurites with secretory granules and NSE [42]. Here, we also report an increase in TUBB3, MAP2, TH and DAT levels during neuroblastoma SH-SY5Y rho^0^ cell differentiation. The common conditions for all these differentiation studies of rho^0^ cells are the presence of high glucose and uridine in the differentiation media. Neuronal differentiation requires the production of large amounts of plasma membrane (plasmalemma). Vulnerable nigral dopaminergic neurons differ from other neurons by having a considerably more complex axonal arborization [1]. Therefore, the formation of axons and dendrites and maintenance of the neuron’s vastly expanded surface require a continuous membrane synthesis [74]. Glycoproteins, glycolipids and phospholipids are major plasmalemma components. Uridine triphosphate (UTP) and glucose are important metabolites in the synthesis of these glycosylated substances. It is possible that the recovery of the capacity of differentiating into dopaminergic neuron after glucose supply in OXPHOS-dysfunctional cells is due to the allocation of glucose to energy-generating glycolysis and to pathways of glycoconjugate synthesis [75]. Moreover, UTP is a precursor of cytidine triphosphate (CTP), which is important in the synthesis of phospholipids [59]. Very interestingly, it has been reported that triacetyluridine, with a greater oral bioavailability than uridine, significantly attenuated MPTP-induced depletion of striatal dopamine and loss of TH-positive neurons in mouse substantia nigra [76]. Potassium-evoked dopamine release was significantly greater among uridine monophosphate (UMP)-treated rats and the levels of neurofilament-70 and neurofilament-N proteins, biomarkers of neurite outgrowth, increased with UMP consumption [58]. UMP treatment reduced ipsilateral rotations and significantly elevated rat striatal dopamine, TH activity and protein, and synapsin-1 protein [77]. Thus, uridine could be used as a protective agent against OXPHOS xenobiotics provided during pregnancy, when dopaminergic neuronal differentiation is taking place. It could also be beneficial during pregnancy of women harboring mtDNA pathologic mutations.

The decreased dopaminergic neurogenesis could be important in the development of PD [18]. Here, we have demonstrated that the exposure to AZT, LIN or ATO reduces dopaminergic neurogenesis. Therefore, early contact with these drugs might be a risk factor for developing PD.

## 5. Conclusions

AZT, LIN and ATO, at concentrations found in human blood, reduce dopaminergic neuronal differentiation of neuroblastoma SH-SY5Y cells and LIN increases global DNA methylation of mouse fetal brains. Uridine inhibits these effects and could be considered during the treatment with these drugs.

## Figures and Tables

**Figure 1 cells-08-01407-f001:**
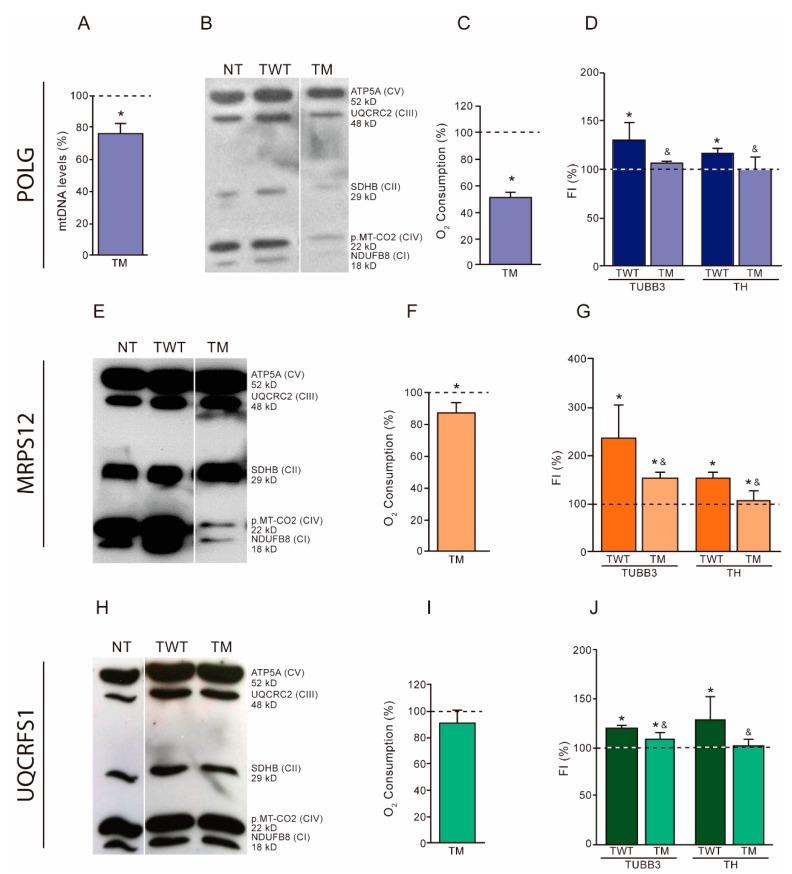
Effects of the overexpression of OXPHOS-related mutant proteins on neuroblastoma SH-SY5Y cells. (**A**–**D**) Wild-type (TWT) or Mutant (TM) *POLG* transfected cells. (**A**) Mitochondrial DNA (mtDNA) levels. Dashed line (100%) represents the mean value of undifferentiated TWT cells. The bar indicates the mean value and standard deviation (independent experiments, N = 4) of undifferentiated TM cells. * *p* < 0.05 (versus TWT cells, Mann–Whitney). (**B**) Levels of representative subunits of OXPHOS complexes. NT, non-transfected cells. ATP5A, complex V (CV) subunit; UQCRC2, complex III (CIII) subunit; SDHB, complex II (CII) subunit; p.MT-CO2, complex IV (CIV) subunit; and NDUFB8, complex I (CI) subunit. Molecular weights are also indicated. (**C**) Oxygen consumption. Dashed line (100%) represents the mean value of undifferentiated TWT cells. The bar indicates the mean value and standard deviation (N = 3) of undifferentiated TM cells. * *p* < 0.05 (versus TWT cells, Mann–Whitney). (**D**) Dopaminergic neuronal differentiation. FI, fluorescence intensity (flow cytometry) of dopaminergic neuronal markers TUBB3 and TH. Dashed line (100%) represents the mean values of undifferentiated TWT or TM cells. Bars indicate the mean values and standard deviations (N = 3) of differentiated TWT or TM cells. * *p* < 0.05 (versus undifferentiated TWT or TM cells, Mann–Whitney). ^&^
*p* < 0.05 (versus differentiated TWT cells, Mann–Whitney). (**E**–**G**) Wild-type (TWT) or Mutant (TM) *MRPS12* transfected cells. (**E**) Levels of representative subunits of OXPHOS complexes. See panel B. (**F**) Oxygen consumption. (N = 3). See panel C. (**G**) Dopaminergic neuronal differentiation. (N = 6). See panel D. (**H**–**J**) Wild-type (TWT) or Mutant (TM) *UQCRFS1* transfected cells. (**H**) Levels of representative subunits of OXPHOS complexes. See panel B. (**I**) Oxygen consumption. (N = 6). See panel C. (**J**) Dopaminergic neuronal differentiation. (N = 3). See panel D.

**Figure 2 cells-08-01407-f002:**
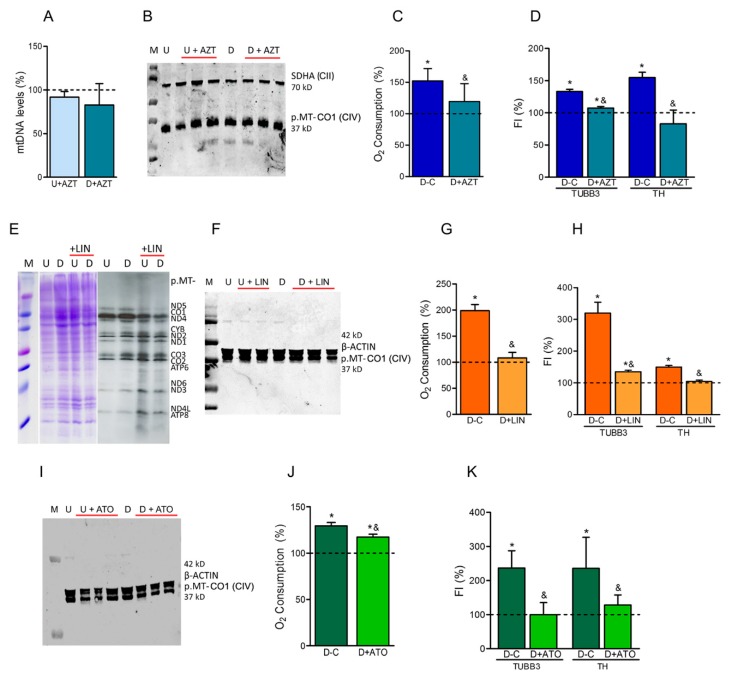
Effects of OXPHOS-xenobiotics on neuroblastoma SH-SY5Y cells. (**A**–**D**) Treatment with azidothymidine (AZT). (**A**) Mitochondrial DNA (mtDNA) levels. Dashed line (100%) represents the mean value of untreated undifferentiated (U) or differentiated (D) cells. Bars indicate the mean values and standard deviations (independent experiments, N = 2) of AZT-treated U or D cells. (**B**) Levels of the p.MT-CO1 subunit from complex IV (CIV) in untreated and AZT-treated (+AZT) cells. The nDNA-encoded complex II (CII) subunit SDHA was used to check the protein loading. Molecular weights are indicated. M, U, D code for molecular weight marker, undifferentiated (U) and differentiated (D) cells, respectively. (N = 3). (**C**) Oxygen consumption. Dashed line (100%) represents the mean value of untreated or treated undifferentiated (U) cells. Bars indicate the mean values and standard deviations (N = 3) of untreated (C) or treated (+AZT) differentiated (D) cells. * *p* < 0.05 (versus U cells, Mann–Whitney). ^&^
*p* < 0.05 (versus untreated D cells, Mann–Whitney). (**D**) Dopaminergic neuronal differentiation. FI, fluorescence intensity (flow cytometry) of dopaminergic neuronal markers TUBB3 and TH. Dashed line (100%) represents the mean value of undifferentiated (U) cells. Bars indicate the mean values and standard deviations (N = 3) of untreated (C) and treated (+AZT) differentiated (D) cells. * *p* < 0.05 (versus U cells, Mann–Whitney). ^&^
*p* < 0.05 (versus untreated D cells, Mann–Whitney). E-H) Treatment with linezolid (LIN). (**E**) Representative image of gels for mitochondrial protein synthesis analysis. Gels showing the loading control and electrophoretic patterns of mitochondrial translation products from untreated and treated (+LIN) cells are included. M, U, D code for molecular weight marker, undifferentiated (U) and differentiated (D) cells, respectively. Proteins p.MT-ND1-6 and p.MT-ND4L are CI mtDNA-encoded subunits; p.MT-CYB is a mtDNA-encoded subunit from CIII; p.MT-CO1-3 are mtDNA-encoded subunits from CIV; and p.MT-ATP6,8 are mtDNA-encoded subunits from CV. (N = 2). (**F**) Levels of p.MT-CO1 subunit in untreated and LIN-treated (+LIN) cells. β-ACTIN was used to check the protein loading. (N = 2). See panel B. (**G**) Oxygen consumption. (N = 3). See panel C. (**H**) Dopaminergic neuronal differentiation. (N = 3). See panel D. (I–K) Treatment with atovaquone (ATO). (**I**) Levels of p.MT-CO1 subunit in untreated and ATO-treated (+ATO) cells. (N = 3). See panels B and F. (**J**) Oxygen consumption. (N = 3). See panel C. (**K**) Dopaminergic neuronal differentiation. (N = 4). See panel D.

**Figure 3 cells-08-01407-f003:**
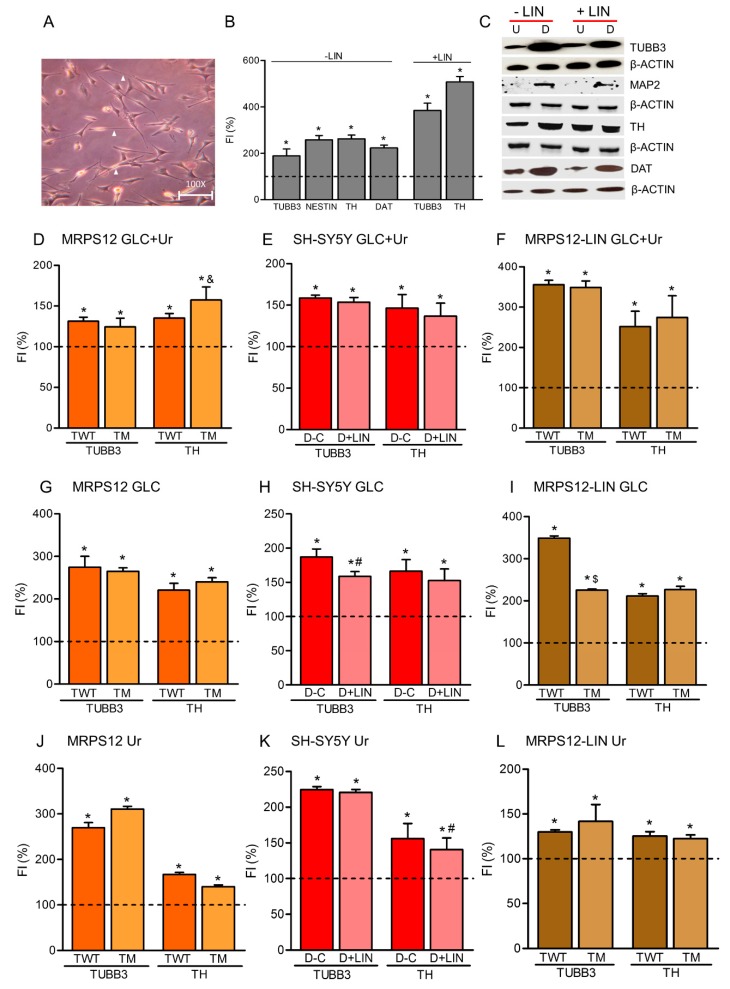
Effects of the high glucose and/or uridine on dopaminergic neuronal differentiation of OXPHOS-dysfunctional cells. (**A**–**C**) Neuroblastoma SH-SY5Y rho^0^ cells. (**A**) Neuron-differentiated rho^0^ cells. Arrowheads indicate neurites. (**B**) Dopaminergic neuronal differentiation with (+LIN) or without (-LIN) linezolid. FI, fluorescence intensity (flow cytometry) of dopaminergic neuronal markers TUBB3, NESTIN, TH and DAT. Dashed line (100%) represents the mean value of undifferentiated (U) rho^0^ cells. Bars indicate the mean values and standard deviations (independent experiments, N = 3) of differentiated (D) rho^0^ cells. * *p* < 0.05 (versus U rho^0^ cells, Mann–Whitney). (**C**) Levels of dopaminergic neuronal markers in untreated (−LIN)- or treated (+LIN)-undifferentiated (U) and differentiated (D) rho^0^ cells. (N = 2). β-ACTIN was used to check the protein loading. (**D**–**F**) Neuronal differentiation of SH-SY5Y rho+ cells in high glucose (GLC) and uridine (Ur) media. (**D**) SH-SY5Y rho^+^ cells overexpressing wild-type (TWT) or mutant (TM) MRPS12 protein. FI, fluorescence intensity (flow cytometry) of dopaminergic neuronal markers TUBB3 and TH. Dashed line (100%) represents the mean value of undifferentiated (U) cells. Bars indicate the mean values and standard deviations (N = 3) of differentiated (D) cells. * *p* < 0.05 (versus U cells, Mann–Whitney). ^&^
*p* < 0.05 (versus TWT D cells, Mann–Whitney). (**E**) Untreated (C) or linezolid treated (+LIN) SH-SY5Y rho^+^ cells. FI, fluorescence intensity (flow cytometry) of dopaminergic neuronal markers TUBB3 and TH. Dashed line (100%) represents the mean value of undifferentiated (U) cells. Bars indicate the mean values and standard deviations (N = 3) of differentiated (D) cells. * *p* < 0.05 (versus U cells, Mann–Whitney). (**F**) Linezolid (LIN)-treated SH-SY5Y rho^+^ cells overexpressing wild-type (TWT) or mutant (TM) MRPS12 protein. FI, fluorescence intensity (flow cytometry) of dopaminergic neuronal markers TUBB3 and TH. Dashed line (100%) represents the mean value of undifferentiated (U) cells. Bars indicate the mean values and standard deviations of differentiated (D) cells. * *p* < 0.05 (versus U cells, Mann–Whitney). G-I) Neuronal differentiation of SH-SY5Y rho+ cells in high glucose (GLC) media. (**G**) SH-SY5Y rho^+^ cells overexpressing wild-type (TWT) or mutant (TM) MRPS12 protein. (N = 3). See panel D. (**H**) Untreated (C) or linezolid treated (+LIN) SH-SY5Y rho^+^ cells. (N = 3). * *p* < 0.05 (versus U cells). ^#^
*p* < 0.05 (versus untreated D cells, Mann–Whitney). See panel E. (**I**) Linezolid (LIN)-treated SH-SY5Y rho^+^ cells overexpressing wild-type (TWT) or mutant (TM) MRPS12 protein. (N = 3). * *p* < 0.05 (versus U cells). ^$^
*p* < 0.05 (versus LIN-treated TWT D cells, Mann–Whitney). See panel F. (**J**–**L**) Neuronal differentiation of SH-SY5Y rho+ cells in high uridine (Ur) media. (**J**) SH-SY5Y rho^+^ cells overexpressing wild-type (TWT) or mutant (TM) MRPS12 protein. (N = 3). See panel D. (K) Untreated (C) or linezolid treated (+LIN) SH-SY5Y rho^+^ cells. (N = 3). See panel E and H. (**L**) Linezolid (LIN)-treated SH-SY5Y rho^+^ cells overexpressing wild-type (TWT) or mutant (TM) MRPS12 protein. (N = 3). See panel F.

**Figure 4 cells-08-01407-f004:**
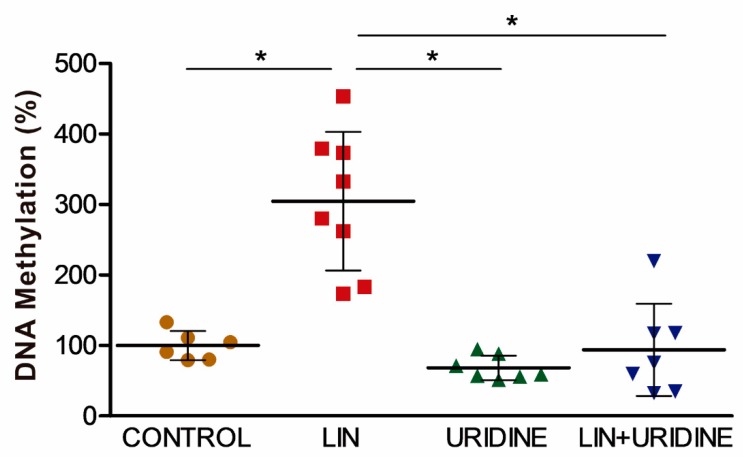
DNA methylation. Points represent individual samples and horizontal lines indicate mean ± standard deviation values. (Independent experiments, N ≥ 6). Kruskal–Wallis, *p* = 0.0004. Horizontal black lines indicate between-treatment *p* values fulfilling the post-hoc Bonferroni/Dunn criteria (* *p* < 0.0083).

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
