# Peer review of "Uridine Prevents Negative Effects of OXPHOS Xenobiotics on Dopaminergic Neuronal Differentiation"

_cells, 2019, doi:10.3390/cells8111407_

Round 1

Reviewer 1 Report

This manuscript mainly focuses on the protective of uridine on OXPHOS xenobiotics caused negative effects on dopaminergic neuronal differentiation. I think the findings are interesting, but some changes are recommended before publication.

General Comments and suggestions:

Uridine is very important agent in this study. However, there is no mention of uridine in the introduction, I recommended adding it. In Materials and Methods, overexpressing OXPHOS-related mutant proteins POLG, MRPS12 and UQCRFS1 proteins were used to study the relation of OXPHOS dysfunction and dopaminergic neuronal differentiation. Please add the reason for choosing these 3 proteins and the definition of POLG, MRPS12 and UQCRFS1. The annotation of figure 1, 2 and 3 was not easy to read, please list the annotation of picture one by one. Moreover, all the abbreviation present in the annotation of figures should be defined. Dopaminergic neuronal differentiation is the most important result, please add the immunohistochemical image to make the result credible which combine with the fluorescence intensity in firure1, 2 and 3. Conclusions need to be more rigorous, such as the “some OXPHOS xenobiotics” can be more specific, please make changes.

Author Response

Comments and Suggestions for Authors

This manuscript mainly focuses on the protective of uridine on OXPHOS xenobiotics caused negative effects on dopaminergic neuronal differentiation. I think the findings are interesting, but some changes are recommended before publication.

            We would like to thank this referee for his/her suggestions.

General Comments and suggestions:

Uridine is very important agent in this study. However, there is no mention of uridine in the introduction, I recommended adding it.

            We have now mentioned it in the “Introduction” section.

In Materials and Methods, overexpressing OXPHOS-related mutant proteins POLG, MRPS12 and UQCRFS1 proteins were used to study the relation of OXPHOS dysfunction and dopaminergic neuronal differentiation. Please add the reason for choosing these 3 proteins and the definition of POLG, MRPS12 and UQCRFS1.

            We chose these OXPHOS-related mutant proteins because they participate in the same mitochondrial processes than that we were going to test with the previously cited OXPHOS xenobiotics (replication - POLG and azidothymidine, translation - MRPS12 and linezolid, and respiratory chain function - UQCRFS1 and atovaquone). We have now added the reason for choosing these proteins in the “Overexpressing OXPHOS-related mutant proteins” section. We have also defined these acronyms in the “Cells, mice and materials” section.

The annotation of figure 1, 2 and 3 was not easy to read, please list the annotation of picture one by one. Moreover, all the abbreviation present in the annotation of figures should be defined.

We have noted that, trying to make the figure legends shorter, we made them more confusing. We have now listed the annotations of the figure panels one by one. All the abbreviations are defined in the figure legends.

Dopaminergic neuronal differentiation is the most important result, please add the immunohistochemical image to make the result credible which combine with the fluorescence intensity in firure1, 2 and 3.

            Fluorescence intensity results in figures 1, 2 and 3 were not obtained by immunohistochemical techniques but by flow cytometry. In any case, we had previously confirmed the validity of this approach by comparing it with the results obtained by Western blot analysis in the “Cell differentiation into dopaminergic neuron” section of Supplemental Note 1.

Conclusions need to be more rigorous, such as the “some OXPHOS xenobiotics” can be more specific, please make changes.

            We have now written the conclusions in a more rigorous way.

Reviewer 2 Report

The manuscript (ms) “Uridine prevents OXPHOS xenobiotics negative effects on dopamminergic neuronal differentiation” results an interesting study and may give a contribution to the scientific community for the effect of environment on neurodegenerative disease. However, there are more questions to be clarified prior to publication.

Major commens

The paper needs to be rewritten because is more confusing.

1) Materials and Methods

The Authors have to separate the materials from the methods. Methods:

The Authors have to delete the methods of supplementary materials, while have to improve the methods inside the paper. Please put methods of supplementary materials in a different file.

Cells culture: Explain the different between rho+ and rho0. Add the stem cell culture condition. Moreover I think that the reference [20] is referred to stem cells.

Overexpressing OXPHOS-related mutant protein: Line 86 there are only acronyms for POLG and MRPS12. The Authors have to specify the kind of cells they used for the overexpressing experiments.

Mice experiments: I think it is better to put at the end of method section. Moreover the Authors have to add two specific paragraphs about RT-PCR and Quantitative detection of DNA methylation.

Analysiss of dopaminergic neuronal differentiation: Again the Authors put different methods together, please add specific paragraphs (i.e. Flow cytometry, WB, Elisa).

Genetic analysis: In this section the Authors wrote about PPAR gamma but they performed RT-PCR so I think there is a mistake.

Analysis of Oxidative phosphorilation function: Please delete, because you did not mentioned it in the ms.

I believe it is more important add a paragraph for treatments in vitro, specifying the vehicle, concentration and time used.

Finally, the Author reports experiments of immunocytochemstry but they did not report which kind of microscopy methods used.

2) Results

OXPHOS function and neuronal differentiation :

Supplemental Note 1: Nestin is a marker of undifferentiated cells why you put it as marker of neuronal differentiation, also DCX in implicated in the migration of neuronal progenitors.

FI%: what does it mean? Is it refered to flow cytometry or WB or other? Explain at least in the legend.

Nestin (NES) increase! It is strange!! How do you explain this result?

N: what does it mean? Explain in all the legends.

You write: “The FC results were confirmed by WB analysis” it is not true for NES

You write “Our differentiation protocol also increased the amount of intracellular dopamine (Figure S4C). Moreover, the differentiated cells were electrically active neurons because they released dopamine after KCl stimulation (Figure S4D)” in the methods section you have to describe better this procedure.

Figure S3 please check typos

Figure S4 B please there are typos in D there is a mistake in the x-axis

Figure S5. Intracellular acetylcholine levels increase. How do you explain this result?

You write:” The peroxisome proliferator-activated receptor gamma (PPARG) has been associated to mitochondrial biogenesis. mRNA levels for this factor significantly increased with the differentiation in the analyzed cells” you have to insert specific reference. Anyway it is true, but there are more specific such as PGC1alpha, NFR2 and all the machinery protein implicated in the import of mitochondrial protein. I think that the data on PPAR g is more interesting, I suggest to deeper investigate this data removing it from the ms this result, why in this manner does not improve it.

OXPHOS dysfunction and dopamminergic neuronal differentiation.

It is in general confused.

At line 228 the Authors write RISP, but they should maintain always the same acronym.

In fig 1A *p TM value is versus TWT the Authors have also to specify versus what *p TWT value is referred. The color of histograms are always different and this do not help the reader. Please revise the figure. Again FI%: what does it mean? Is it refered to flow cytometry or WB or other? Explain at least in the legend.

OXPHOS xenobiotics and dopamminergic neuronal differentiation

It is in general confused. If you speak about protein never mentioned before please do not put the acronym ie P.MT-CO1.

It is redundant speak about the effects of these xenobiotics in the results section, please remove.

Moreover you have to explain the choice of concentration in the material and methods section and not in the results. Moreover I understand that is the first time that these substances are used on shsy5y cells differentiated and not. Therefore the Author have to perform a viability assay with the concentration choice.

In Fig 2 I,J, K are absent.

In the legend of fig 2 the Authors have to write extended acronym of the protein bands.

The colors of histograms are always different and this does not help the reader.

Neuroblastoma SH-SY5Y rho0 cell line

The SH-SY5Y rho0 cell line differentiated towards dopamminergic phenotype; my question is: does these cells differentiate as well as SH-SY5Y rho+? I think this is an important parameter to valuate. In absence of mitochondrial DNA they Authors did not observe the adverse effect of LIN on neuronal maturation; am I right? Sincerely I do not understand the section! In fig3B you again report Nestin (NSC marker).

Rho0 medium recovers dopamminergic neuronal differentiation of OXPHOS-dysfunctional cells.

It seems that the Authors use a conditioned medium.

FI%: what does it mean? Is it refered to flow cytometry or WB or other? Explain at least in the legend.

The Authors saw hypermethylation in mice why did they not perform this analysis also in cell culture?

Minor comments

There are many typos, please check the ms: i.e. line 79 instead (instead of); line 118either “with” vehicle ….or….; line 182 analyses (analysis)

Author Response

Comments and Suggestions for Authors

The manuscript (ms) “Uridine prevents OXPHOS xenobiotics negative effects on dopamminergic neuronal differentiation” results an interesting study and may give a contribution to the scientific community for the effect of environment on neurodegenerative disease.

            We would like to thank this referee for his/her supporting comments and extensive review, which will improve our manuscript.

However, there are more questions to be clarified prior to publication.

Major commens
The paper needs to be rewritten because is more confusing.

            We have rewritten the confusing sections of the paper.

1) Materials and Methods

The Authors have to separate the materials from the methods.

            We have separated the materials from the methods.

Methods:

The Authors have to delete the methods of supplementary materials, while have to improve the methods inside the paper. Please put methods of supplementary materials in a different file.

            To avoid the repetition of some methods in the main text and supplementary materials, we have maintained all the methods together in the main text, but we have improved them, according to this referee’s suggestions.

Cells culture: Explain the different between rho+ and rho0. Add the stem cell culture condition. Moreover I think that the reference [20] is referred to stem cells.

            Rho+ and rho0 cells are cells with or without mtDNA, respectively. We have now explained the difference between these cells in the first paragraph of the “Cells, mice and materials” section.

We have also added the culture conditions for the human neural stem cells (hNSCs) and specified that reference [20], now [22], refers to dopaminergic differentiation of stem cells in the “Cell culture and differentiation” section.

Overexpressing OXPHOS-related mutant protein: Line 86 there are only acronyms for POLG and MRPS12. The Authors have to specify the kind of cells they used for the overexpressing experiments.

            We have now given the names for these proteins (POLG- mitochondrial DNA polymerase gamma; and MRPS12- mitochondrial ribosomal protein S12) in the “Cells, mice and materials” section.

            The cells used in the overexpressing experiments were neuroblastoma SH-SY5Y cells. This is now specified in the text (“Overexpressing OXPHOS-related mutant proteins” section).

Mice experiments: I think it is better to put at the end of method section. Moreover the Authors have to add two specific paragraphs about RT-PCR and Quantitative detection of DNA methylation.

            We have now transferred this section to the end of methods, just before of the “Statistics analysis” section. We have also added two specific sections (“RT-qPCR analysis” and “DNA methylation analysis”) in Materials and Methods.

Analysiss of dopaminergic neuronal differentiation: Again the Authors put different methods together, please add specific paragraphs (i.e. Flow cytometry, WB, Elisa).

            We have now added specific sections for the flow cytometry, Western blotting and ELISA methods.

Genetic analysis: In this section the Authors wrote about PPAR gamma but they performed RT-PCR so I think there is a mistake.

            We have transferred the PPAR gamma analysis to the new “RT-qPCR analysis” section and left, in this section, the references or protocols to determine the nuclear genetic fingerprint, the karyotyping, the mtDNA levels and mtDNA sequencing of SH-SY5Y cells. However, this section is now named “Chromosomes and mitochondrial DNA analysis”.

Analysis of Oxidative phosphorilation function: Please delete, because you did not mentioned it in the ms.

            Several OXPHOS-related variables, such as mtDNA levels, mitochondrial translation, subunits of respiratory complexes and, mainly, oxygen consumption are mentioned in figures 1 and 2 legends.

I believe it is more important add a paragraph for treatments in vitro, specifying the vehicle, concentration and time used.

            We have now added a specific section (“Pharmacological treatments”) for the in vitro treatments, indicating the vehicle, concentration and time used.

Finally, the Author reports experiments of immunocytochemstry but they did not report which kind of microscopy methods used.

            We have added an “Immunocytochemistry” section to Materials and Methods and, there, we have specified the microscopy methods used.

2) Results
OXPHOS function and neuronal differentiation :

Supplemental Note 1: Nestin is a marker of undifferentiated cells why you put it as marker of neuronal differentiation, also DCX in implicated in the migration of neuronal progenitors.

Nestin (NES) is an intermediate filament marker detected in neural progenitor cells 1. Its levels should decrease along neuronal differentiation 2. As the referee comments, DCX is involved in the migration of neuronal progenitors. These markers are frequently used to check neuronal differentiation of hNSCs and, along with TUBB3 and MAP2, have also been tested in neuroblastoma SH-SY5Y cells 3,4. Therefore, to analyze the neuronal differentiation of neuroblastoma SH-SY5Y cells, we compared all these markers to those of hNSCs.

1 Wiese et al, 2004. Nestin expression – a property of multi-lineage progenitor cells? Cell Mol Life Sci 61: 2510-22.

2 Bazan et al, 2004. In vitro and in vivo characterization of neural stem cells. Histol Histopathol 19: 1261-75.

3 Lopes et al, 2010. Comparison between proliferative and neuron-like SH-SY5Y cells as an in vitro model for Parkinson diseases studies. Brain Res 1337: 85-94.

4 Turaç et al, 2013. Combined flow cytometric analysis of surface and intracellular antigens reveals surface molecule markers of human neuropoiesis. PLoS One 8: e68519.

FI%: what does it mean? Is it refered to flow cytometry or WB or other? Explain at least in the legend.

FI % means fluorescence intensity referred to flow cytometry. We have noted that, trying to make the figure legends shorter, we made them more confusing. We have now corrected this problem.

Nestin (NES) increase! It is strange!! How do you explain this result?

Yes, NES increased in differentiated cultures of SH-SY5Y cells, despite being a marker of neural progenitor cells. We also expected a decrease along neuronal differentiation, as we observed for hNSCs. This was strange, but we found another article reporting the same fact 1. Our morphological and biochemical data suggest that SH-SY5Y cell populations at different stages of differentiation are represented in differentiated cultures: undifferentiated, neuroblastoma tumor-like cells; partially differentiated stem-like, NES-positive cells; and neuron-like cells with elongated neurites and expressing mature neuronal markers (TUBB3, NSE, MAP2). Such asynchronous differentiation has been also reported in another experimental SH-SY5Y model 2. Perhaps, in the differentiated cultures of SH-SY5Y cells, the number of partially differentiated stem-like, NES-positive cells surpasses the number of undifferentiated, neuroblastoma tumor-like cells. We have added this potential explanation to the text.

1 Pezzini et al, 2017. Transcriptomic profiling discloses molecular and celular events related to neuronal differentiation in SH-SY5Y neuroblastoma cells. Cell Mol Neurobiol 37: 665-82.

2 da Rocha et al, 2015. Analysis of the amyloid precursor protein role in neuritogénesis reveals a biphasic SH-SY5y neuronal cell differentiation model. J Neurochem 134: 288-301.

N: what does it mean? Explain in all the legends.

            N is the number of independent experiments. We have now explained it in all the figure legends.

You write: “The FC results were confirmed by WB analysis” it is not true for NES

            As the referee indicates, flow cytometry (FC) results for NES were not confirmed by Western blot (WB). However, when we wrote: “Flow cytometry (FC) analysis shows that NES levels decrease in hNSCs. However, and similar to other report [22], they increase in the SH-SY5Y cell line after our differentiation protocol (Figure S3A). TUBB3 and DCX levels increase after the differentiation in both neuroblastoma cell line and hNSCs. Western blot (WB) analyses confirm the increase of these, and 2 other, neural markers, such as MAP2 and NSE (Figure S3B)”, the “these” term in the last sentence was referred to TUBB3 and DCX. This has been clarified now in the text.

You write “Our differentiation protocol also increased the amount of intracellular dopamine (Figure S4C). Moreover, the differentiated cells were electrically active neurons because they released dopamine after KCl stimulation (Figure S4D)” in the methods section you have to describe better this procedure.

            We have now better described this procedure in the methods section.

Figure S3 please check typos

            We have checked them.

Figure S4 B please there are typos in D there is a mistake in the x-axis

            We have corrected the typos and the x-axis mistake.

Figure S5. Intracellular acetylcholine levels increase. How do you explain this result?

            The intracellular acetylcholine levels do not increase. In both differentiated hNSCs and differentiated SH-SY5Y cells (bars), intracellular acetylcholine levels do not differ from those of undifferentiated cells (dashed line). In fact, they tend to be lower.

You write:” The peroxisome proliferator-activated receptor gamma (PPARG) has been associated to mitochondrial biogenesis. mRNA levels for this factor significantly increased with the differentiation in the analyzed cells” you have to insert specific reference. Anyway it is true, but there are more specific such as PGC1alpha, NFR2 and all the machinery protein implicated in the import of mitochondrial protein. I think that the data on PPAR g is more interesting, I suggest to deeper investigate this data removing it from the ms this result, why in this manner does not improve it.

            We have added a specific reference 1.

This is a very interesting suggestion and we will investigate the PGC1alpha and NRF2 expression in the future to confirm the mitochondrial biogenesis during dopaminergic neuronal differentiation.

1 Miglio et al, 2009. PPARgamma stimulation promotes mitochondrial biogenesis and prevents glucose deprivation-induced neuronal loss. Neurochem Int 55: 496-504.

OXPHOS dysfunction and dopamminergic neuronal differentiation.

It is in general confused.

At line 228 the Authors write RISP, but they should maintain always the same acronym.

            We have now changed RISP by UQCRFS1.

In fig 1A *p TM value is versus TWT the Authors have also to specify versus what *p TWT value is referred.

Panels A to D in Figure 1 refer to cells transfected with mutant or wild-type POLG sequences. Here (Figure 1A), TWT means cells transfected with POLG wild-type sequence. We have now made the figure and figure legend clearer.

The color of histograms are always different and this do not help the reader. Please revise the figure. Again FI%: what does it mean? Is it refered to flow cytometry or WB or other? Explain at least in the legend.

            We chose these OXPHOS-related mutant proteins because they participate in the same mitochondrial processes that we were going to test with OXPHOS xenobiotics (replication - POLG and azidothymidine; translation - MRPS12 and linezolid; and respiratory chain function - UQCRFS1 and atovaquone). We decided to give a color code for similar processes (bluish for mitochondrial replication; reddish for mitochondrial translation; and greenish for respiratory chain function). We have added labels to clarify the gene overexpressed in each case.

FI % means fluorescence intensity referred to flow cytometry. We have now explained it in the figure legend.

OXPHOS xenobiotics and dopamminergic neuronal differentiation

It is in general confused. If you speak about protein never mentioned before please do not put the acronym ie P.MT-CO1.

            p.MT-CO1 refers to subunit I of the respiratory complex IV (cytochrome oxidase). We have now added the name before the acronym in the “Cells, mice and materials” section.

It is redundant speak about the effects of these xenobiotics in the results section, please remove.

Moreover you have to explain the choice of concentration in the material and methods section and not in the results. Moreover I understand that is the first time that these substances are used on shsy5y cells differentiated and not. Therefore the Author have to perform a viability assay with the concentration choice.

            As previously suggested by this referee, we have now added, in “Materials and Methods”, a section for in vitro treatments, specifying the vehicle, concentration and time used. The comments on the effects of these xenobiotics have been deleted from the Results and transferred to Materials and Methods.

            We performed viability assays. SH-SY5Y cells were seeded and grown for a week with or without OXPHOS xenobiotics at the reported concentrations. The protein concentrations of untreated (U) and treated (T) cells were determined [5 μM AZT, U = 8.7 mg/ml ± 2.7 (4), T = 8.9 mg/ml ± 1.7 (4); 40 μM LIN, U = 11.3 mg/ml ± 5.2 (4), T = 12.1 mg/ml ± 8.3 (4); 0.5 μM ATO, U = 9.3 mg/ml ± 2.0 (4), T = 10.9 mg/ml ± 5.8+ (4)] and there were not significant differences. This is now referred in the manuscript. Moreover, similar or higher drug concentrations had been previously tested in other cell lines and viability was not significantly altered 1-9. Finally, the xenobiotics concentrations that we used in cell culture are those reported in human blood 10-12.

1 Pan-Zhou et al, 2000. Differential effects of antiretroviral nucleoside analogs on mitochondrial function in HepG2 cells. Antimicrob Agents Chemother 44: 496-503.

2 Liu et al, 2012. Molecular analysis of mitochondrial compromise in rodent cardiomyocytes exposed long term to nucleoside reverse transcriptase inhibitors (NRTIs). Cardiovasc Toxicol 12: 123-34.

3 Hung et al, 2017. Mitochondrial defects arise from nucleoside/nucleotide reverse transcriptase inhibitors in neurons: potential contribution to HIV-associated neurocognitive disorders. Biochim Biophys Acta Mol Basis Dis 1863: 406-13.

4 Duewelhenke et al, 2007. Influence on mitochondrial and cytotoxicity of different antibiotics administered in high concentrations on primary human osteoblasts and cell lines. Antimicrob Agents Chemother 51: 54-63.

5 Fiorillo et al, 2016. Repurposing atovaquone: targeting mitochondrial complex III and OXPHOS to eradicate cancer stem cells. Oncotarget 7: 34084-99.

6 Chen et al, 2018. Targeting mitochondria by anthelmintic drug atovaquone sensitizes renal cell carcinoma to chemotherapy and immunotherapy. J Biochem Mol Toxicol 32: e22195.

7 Lv et al, 2018. Atovaquone enhances doxorubicin’s efficacy via inhibiting mitochondrial respiration and STAT3 in aggressive thyroid cancer. J Bioenerg Biomembr 50: 263-70.

8 Tian et al, 2018. Targeting mitochondrial respiration as a therapeutic strategy for cervical cancer. Biochem Biophys Res Commun 499: 1019-24.

9 Sun 2019. Inhibition of mitochondrial respiration overcomes hepatocellular carcinoma chemoresistance. Biochem Biophys Res Commun 508: 626-32.

10 Demir et al, 2015. Neurotoxic effects of AZT on developing and adult neurogenesis. Front Neurosci 9: 93.

11 Dryden, 2013. Linezolid pharmacokinetics and pharmacodynamics in clinical treatment. J Antimicrob Chemother 66: iv7-iv15.

12 Calderon et al, 2016. Efavirenz but not atazanavir/ritonavir significantly reduces atovaquone concentrations in HIV-infected subjects. Clin Infect Dis 62: 1036-42.

In Fig 2 I,J, K are absent.

They were present in the second line of the figure legend, but as previously mentioned, we have noted that the figure legends were not correctly explained. We have now tried to make them clearer.

In the legend of fig 2 the Authors have to write extended acronym of the protein bands.

The colors of histograms are always different and this does not help the reader.

            We have now indicated the name of all these mtDNA-encoded (subunits 1 to 6 and 4L of NADH:ubiquinone oxidoreductase, respiratory complex I; subunit CYB of cytochrome bc1, respiratory complex III; subunits 1 to 3 of cytochrome oxidase, respiratory complex IV; subunits 6 and 8 of ATP synthase, OXPHOS complex V) and nDNA-encoded (subunit A of succinate dehydrogenase, respiratory complex II) proteins.

            As previously commented, the color of the histograms groups different mitochondrial processes, such as replication (POLG and AZT, bluish histograms); translation (MRPS12 and LIN, reddish histograms); and respiratory chain function (UQCRFS1 and ATO, greenish histograms).

Neuroblastoma SH-SY5Y rho0 cell line

The SH-SY5Y rho0 cell line differentiated towards dopamminergic phenotype; my question is: does these cells differentiate as well as SH-SY5Y rho+? I think this is an important parameter to valuate.

            Yes, when grown with high glucose and/or uridine, cells differentiate as well as rho+ cells in normal differentiation medium, as it can be observed when comparing results of figure 3B and C with those of figures S3A and B or S4A and B.

In absence of mitochondrial DNA they Authors did not observe the adverse effect of LIN on neuronal maturation; am I right? Sincerely I do not understand the section!

            Yes, you are right. In absence of mitochondrial DNA, the adverse effect of LIN on neuronal maturation is not observed.

Among other functions, OXPHOS is involved in ATP and uridine production 1. LIN reduces mitochondrial translation and, therefore, it will have a negative effect on OXPHOS function, on ATP and uridine production and on neuronal maturation. As they do not have OXPHOS function, rho0 cells require high glucose and uridine to survive 2. They are always grown in rho0 medium (high glucose and uridine). As the medium is providing glucose and uridine, OXPHOS function is not longer required for neuronal maturation and, therefore, LIN has no effect.

1 Pesini et al, 2019. Brain pyrimidine nucleotide synthesis and Alzheimer disease. Aging (Albany NY) doi: 10.18632/aging.102328 [Epub ahead of print].

2 King et al, 1989. Human cells lacking mtDNA: repopulation with exogenous mitochondria by complementation. Science 246: 500-3.

In fig3B you again report Nestin (NSC marker).

            Yes, we wanted to compare the neuronal differentiation of rho+ and rho0 cells.

Rho0 medium recovers dopamminergic neuronal differentiation of OXPHOS-dysfunctional cells.

It seems that the Authors use a conditioned medium.

            Rho0 medium is not a conditioned medium. As previously explained, this medium contains high glucose and uridine because these cells require higher concentrations of these two compounds to survive.

FI%: what does it mean? Is it refered to flow cytometry or WB or other? Explain at least in the legend.

FI % means fluorescence intensity referred to flow cytometry. We have now explained it in the figure legend.

The Authors saw hypermethylation in mice why did they not perform this analysis also in cell culture?

            We observed a direct effect of all these OXPHOS xenobiotics on neuronal differentiation of SH-SY5Y cells, but we did not note it in mice fetuses. This result made us to think in a potential delayed effect 1, and DNA methylation, as an epigenetic marker, was a good candidate. In any case, after our result in mice, this parameter will be considered for the next project.

1 Iglesias et al, 2018. Prenatal exposure to oxidative phosphorylation xenobiotics and late-onset Parkinson disease. Ageing Res Rev 45: 24-32.

Minor comments

There are many typos, please check the ms: i.e. line 79 instead (instead of); line 118either “with” vehicle ….or….; line 182 analyses (analysis)

            We have now corrected these typos.

Reviewer 3 Report

This manuscript appears to be rather confused and presents at best only weak evidence in support of its conclusion. In general it needs to decide on what it is trying to say and build a case based on te data shown. 

Major points:

1. It begins with a first line stating "Neuronal differentiation appears to be dependent on oxidative phosphorylation capacity" yet in section 3.4 they shows that SH-SY5Y rho cells with no OxPhos capcity can indeed differentiate into dopamine-like cells. Don't these contradict each other? 

2.  SH-SY5Y cells may be able to differentiate into neuronal like cells, but they are not neurons. Therefore can the conclusions from these cell studies really be extrapolated to neurons?  In fact they fail to show an effect on neurodifferentiation in the mouse brain.

3. Throughout the text, there are jumps between results of experiments actually carried out and those in other papers. This is very confusing.

4. The abstract fails to explain the rationale for the experiments and what question is being addressed. Then the introduction starts with a discussion of PD. What is ther purpose of this work to look at the OxPhos role in prenatal neuodevelopmental disorders or is it addressing a question about risk fsactors for PD? I note that there is no mention of PD in the discussion.

5. I find the data in most figures unconvincing and patchy. The data for differentiation is a good case, although Fig 1G and Fig 2H show what appears to be clear differences, the rest of the figures show no convincing differences. Howver, I note that the statistical analysis is based on the difference between undifferentiated v differentiated cells.This is not a good reference point, and surely we need to see the statistical significance against the untreated or WT controls. I also note that the Figure legends are not very clear.

6. Although the authors see no overt difference in neuronal differentiation, they see a DNA methylation change in the fetal mouse brain. However this could mean anything - and there is no real reason to link it to neuronal differentiation. It is an overinterpretation to extrapolate this observation to an epeigentic change leading to increse risk for PD later in life. 

7. The conclusions are actually summarised in two lines on the last page. This  is indicative of the low information content of this manuscript. However, this are actiually a summary of the observations, not a conclusion at all. In fact if I was to draw a conclusion from the data presented, I would be more interested in the lack of effect of the drugs seen in SH-SY5Y cells in the absence OxPhos. Not only does this serve as a good control, the result suggests to me that OxPhos may in fact be a modifiying factor that converges the drugs into the biologically active form.

Author Response

Comments and Suggestions for Authors

This manuscript appears to be rather confused and presents at best only weak evidence in support of its conclusion. In general it needs to decide on what it is trying to say and build a case based on te data shown. 

Major points:

It begins with a first line stating "Neuronal differentiation appears to be dependent on oxidative phosphorylation capacity" yet in section 3.4 they shows that SH-SY5Y rho cells with no OxPhos capcity can indeed differentiate into dopamine-like cells. Don't these contradict each other? 

Among other functions, OXPHOS is involved in ATP and uridine production 1. As they do not have OXPHOS function, rho0 cells require high glucose and uridine to survive and proliferate 2. Rho0 cells are grown in rho0 media (high glucose and uridine). One of our findings is that these compounds can replace the OXPHOS function for neuronal differentiation. In normal media, without high glucose and uridine, OXPHOS function will be necessary to provide enough energy and uridine. We have made this issue clearer in the manuscript.

1 Pesini et al, 2019. Brain pyrimidine nucleotide synthesis and Alzheimer disease. Aging (Albany NY) doi: 10.18632/aging.102328 [Epub ahead of print].

2 King et al, 1989. Human cells lacking mtDNA: repopulation with exogenous mitochondria by complementation. Science 246: 500-3.

 SH-SY5Y cells may be able to differentiate into neuronal like cells, but they are not neurons. Therefore can the conclusions from these cell studies really be extrapolated to neurons?  In fact they fail to show an effect on neurodifferentiation in the mouse brain.

            The nervous system is comprised of a vast diversity of distinct neuron types 1,2. Similar to SH-SY5Y cells, human neural stem cells (hNSCs) are also able to in vitro differentiate to neuron-like cells, but are they real neurons? As we comment in the Supplemental Note 1, “The neuroblastoma SH-SY5Y cell line has been frequently used for PD study [1, 2]. In fact, by considering altogether ‘SH-SY5Y’ and ‘Parkinson’s’ terms, more than 1,300 publications appear in the PubMed database”. This cell line has become a popular cell model for PD research because it owns many characteristics of dopaminergic neurons 3-5, similar to those obtained after the neuronal differentiation of hNSCs, as we show in the “Cell differentiation into dopaminergic neuron” section of our Supplemental Note 1. Based on bibliographic data, it appears that neuroblastoma SH-SY5Y cell line is a convenient and useful model.

1 Lodato et al, 2015. Generating neuronal diversity in the mammalian cerebral cortex. Annu Rev Cell Dev Biol 31: 699-720.

2 Tsunemoto et al, 2015. Forward engineering neuronal diversity using direct reprogramming. EMBO J 34: 1445-55.

3 Påhlman et al, 1990. Human neuroblastoma cells in culture: a model for neuronal cell differentiation and function. Acta Physiol Scand Suppl 592: 25-37.

4 Xie et al, 2010. SH-SY5Y human neuroblastoma cell line: in vitro cell model of dopaminergic neurons in Parkinson’s disease. Chin Med J (Engl) 123: 1086-92.

5 Xicoy et al, 2017. The SH-SY5Y cell line in Parkinson’s disease research: a systematic review. Mol Neurodegener 12: 10.

Throughout the text, there are jumps between results of experiments actually carried out and those in other papers. This is very confusing.

            We have now corrected these jumps.

The abstract fails to explain the rationale for the experiments and what question is being addressed. Then the introduction starts with a discussion of PD. What is ther purpose of this work to look at the OxPhos role in prenatal neuodevelopmental disorders or is it addressing a question about risk fsactors for PD? I note that there is no mention of PD in the discussion.

            In the abstract, we commented, “Several therapeutic drugs inhibit oxidative phosphorylation and might be detrimental for neuronal differentiation... To analyze a potential negative effect of three widely used medicaments, we studied the in vitro dopaminergic neuronal differentiation of cells exposed to pharmacologic concentrations of these compounds…”. This is the question that is being addressed. Just before, we had indicated, “Neuronal differentiation appears to be dependent on the oxidative phosphorylation capacity”. This is the rationale for the experiments.

In the introduction, we first comment the OXPHOS function role on late-onset Parkinson disease (PD) 1. The question that we address is on OXPHOS xenobiotics as risk factors for PD. As a decreased neurogenesis has been proposed as a key player for PD 2, and we are able to confirm that OXPHOS xenobiotics affect neuronal differentiation, in the last paragraph of the introduction, we hypothesize that early exposure to OXPHOS xenobiotics might be a risk factor for developing PD 3. We have now incorporated a short paragraph in the “Discussion” section relating our findings to a potential increase in the risk of suffering PD.

1 López-Gallardo et al, 2011. OXPHOS toxicogenomics and Parkinson’s disease. Mutat Res 728: 98-106.

2 Le Grand et al, 2015. Neural stem cells in Parkinson’s disease: a role for neurogenesis defects in onset and progression. Cell Mol Life Sci 72: 773-97.

3 Iglesias et al, 2018. Prenatal exposure to oxidative phosphorylation xenobiotics and late-onset Parkinson disease. Ageing Res Rev 45: 24-32.

I find the data in most figures unconvincing and patchy. The data for differentiation is a good case, although Fig 1G and Fig 2H show what appears to be clear differences, the rest of the figures show no convincing differences. Howver, I note that the statistical analysis is based on the difference between undifferentiated v differentiated cells.This is not a good reference point, and surely we need to see the statistical significance against the untreated or WT controls. I also note that the Figure legends are not very clear.

Our results are based in statistical analysis (non-parametric Mann-Whitney U test) of independent experiments. We do not interpret if the differences are big or small, only we consider if they are statistically significant or not. In several cases, the results of flow cytometry studies were confirmed by Western blot (compare Figure 3B to 3C; compare Figure S3A to S3B; compare Figure S4A and S4B) or immunocytochemistry (compare Figure S3A and S4A to Figure S5A and S5B) analysis. If the term “unconvincing data” means small differences, despite of being statistically significant, the more “unconvincing” results are the more expected ones, those found in cells overexpressing mutant proteins or treated with drugs, where significant differences were not expected. On the other hand, in most of the figures, the differences are very big (for example, Figures 1B, 1C, 1E, 2G, 2K, 3B, 3F, 3G, 3I, 4, S4C, S4D, S6A, S7, S8, S9, S10).

Undifferentiated versus differentiated cells (see * in Figure 1G and 2H), wild-type versus mutant cells (see & in Figure 1G) and untreated versus treated cells (see & in Figure 2H) were compared and, when significant differences were found, they were indicated by different signs (*, &, #, $).

The referee is right about the figure legends. We have noted that trying to make the figure legends shorter, we made them more confusing. We have now made them clearer.

Although the authors see no overt difference in neuronal differentiation, they see a DNA methylation change in the fetal mouse brain. However this could mean anything - and there is no real reason to link it to neuronal differentiation. It is an overinterpretation to extrapolate this observation to an epeigentic change leading to increse risk for PD later in life. 

            We are not relating differences in DNA methylation with differences in neuronal differentiation. We only linked linezolid treatment with differences in DNA methylation and suggested that “if OXPHOS-xenobiotics prenatal toxicity is involved in PD, this early exposure should interfere with developmental programming of nigrostriatal neurons [62]. Thus, the long-lasting modification of global DNA methylation levels might be the connection between these two distant life events: prenatal exposure and old age PD manifestation”.

The conclusions are actually summarised in two lines on the last page. This  is indicative of the low information content of this manuscript. However, this are actiually a summary of the observations, not a conclusion at all. In fact if I was to draw a conclusion from the data presented, I would be more interested in the lack of effect of the drugs seen in SH-SY5Y cells in the absence OxPhos. Not only does this serve as a good control, the result suggests to me that OxPhos may in fact be a modifiying factor that converges the drugs into the biologically active form.

            In the “Discussion” section, we have analyzed our own data, compared them with data from other publications and speculated on the importance of our results. However, the “Conclusions” section sums up the confirmed important results, without any speculation, as is generally required.

As we have responded to his/her first comment, among other functions, OXPHOS is involved in energy and uridine production 1. Linezolid reduces mitochondrial translation and, therefore, it will have a negative effect on OXPHOS function, on energy and uridine production and on neuronal maturation. As they do not have OXPHOS function, rho0 cells require high glucose and uridine to survive 2. They are always grown in rho0 medium (high glucose and uridine). As the medium is providing glucose and uridine, OXPHOS function is not longer required for neuronal maturation and, therefore, linezolid has no effect.

1 Pesini et al, 2019. Brain pyrimidine nucleotide synthesis and Alzheimer disease. Aging (Albany NY) doi: 10.18632/aging.102328 [Epub ahead of print].

2 King et al, 1989. Human cells lacking mtDNA: repopulation with exogenous mitochondria by complementation. Science 246: 500-3.

Round 2

Reviewer 2 Report

Now the manuscript is readable and best organized. The Authors have done a very good work anyway there are  some spelling errors, moreover I am not convinced about the method choose for test the cell viability.  You must use Trypan Blue, MTS, LDH or Neutral red assay but not the protein amount. I think  the Authors have to delete the sentence in lane 129 and perform one of the assay suggested.

Author Response

Now the manuscript is readable and best organized. The Authors have done a very good work

            We consider that his/her previous suggestions largely helped to improve the manuscript. Thank you very much for that comprehensive review.

anyway there are  some spelling errors,

            We have now re-reviewed the manuscript and corrected these errors.

moreover I am not convinced about the method choose for test the cell viability.  You must use Trypan Blue, MTS, LDH or Neutral red assay but not the protein amount. I think  the Authors have to delete the sentence in lane 129 and perform one of the assay suggested.

            In line 129, we wrote “of 0.8 μM and peak concentrations (Cmax) of around 5 μM AZT [24]”. This sentence is not related to cell viability. We suppose that referee is referring to sentence in lane 149 “Cell viability”.

A plethora of tests that assess cell viability are available, however, suffer from a large number of limitations 1,2. It has been reported that: “Cellular viability represents the number of healthy cells present in a population. Cellular proliferation represents the ability of healthy cells to divide and create progeny” 3. Dead cells are detached from the culture plates and lost when the culture media are changed. It was also commented that: “Generally, methods used to determine viability are also common for the detection of cell proliferation” 4. In fact, some kits for cell proliferation are based in some of the assays suggested by the referee [The CellTiter96® Aqueous One Solution Cell Proliferation Assay System Protocol (from Promega G3580) is based in MTS; The Neutral Red Cell Proliferation and Cytotoxicity Assay Kit (from Boster AR1157) is based in neutral red assay]. Moreover, in other articles, the cell viability assay was based in the measurement of cellular protein content using the sulphorhodamine (SRB) assay 5. According to the SRB Assay/Sulforhodamine B Assay Kit (from Abcam ab235935), the measurement of the number of cells in the assay allows its use for analysis of cell viability, cytotoxicity and cell proliferation.

We showed that AZT, LIN and ATO did not affect cell proliferation, determined by protein concentration measurements. In other cell lines, similar concentrations to those that we use did not affect cell viability when determined by some of the assays suggested by the referee 5-7.

            However, to be more rigorous, we have changed “Cell viability” in lane 149 by “Cell proliferation” and, in any case, If the reviewer considers that this is an essential issue, and the editor provides us with at least 1 month (to buy and receive the kit and perform the experiments), we would be happy to do so. We cannot do any of the suggested viability tests in the five days given to us to answer the comments of this referee.

1 Riss et al, 2016. Cell viability assays. In: Sittampalam GS, Grossman A, Brimacombe K, Arkin M, Auld D, Austin C, Baell J, Bejcek B, Caaveiro JMM, Chung TDY, Coussens NP, Dahlin JL, Devanaryan V, Foley TL, Glicksman M, Hall MD, Haas JV, Hoare SRJ, Inglese J, Iversen PW, Kahl SD, Kales SC, Kirshner S, Lal-Nag M, Li Z, McGee J, McManus O, Riss T, Trask OJ Jr., Weidner JR, Wildey MJ, Xia M, Xu X, editors. Assay Guidance Manual [Internet]. Bethesda (MD): Eli Lilly & Company and the National Center for Advancing Translational Sciences; 2004-2013 May 1 [update 2016 Jul 1]. PMID: 23805433.

2 Halim AB, 2018. Do we have a satisfactory cell viability assay? Review of the currently commercially-available assays. Curr Drug Discov Technol. Sep 24. doi: 10.2174/1570163815666180925095433. [Epub ahead of print].

3 Gordon et al, 2018. Cell-based methods for determination of efficacy for candidate therapeutics in the clinical management of cancer. Diseases. Sep 22, 6(4). Pii:E85. doi: 10.3390/diseases6040085. PMID: 30249005.

4 Adan et al, 2016. Cell proliferation and cytotoxicity assays. Curr Pharm Biotechnol. 17(14):1213-1221.

5 Fiorillo et al, 2016. Repurposing atovaquone: targeting mitochondrial complex III and OXPHOS to eradicate cancer stem cells. Oncotarget 7: 34084-99.

6 Hung et al, 2017. Mitochondrial defects arise from nucleoside/nucleotide reverse transcriptase inhibitors in neurons: potential contribution to HIV-associated neurocognitive disorders. Biochim Biophys Acta Mol Basis Dis 1863: 406-13.

7 Duewelhenke et al, 2007. Influence on mitochondrial and cytotoxicity of different antibiotics administered in high concentrations on primary human osteoblasts and cell lines. Antimicrob Agents Chemother 51: 54-63.